

# Prosaposin in seminal plasma on the day of oocyte retrieval is associated with normal fertilization and embryo development in in vitro fertilization cycles

Chun Xu[1], Jiali Cai[1,2], Lanlan Liu[1,2] and Jianzhi Ren[1]

[1] Reproductive Medicine Centre, Chinese PLA 174th Hospital, Xiamen, Fujian, China
[2] Medical College, Xiamen University, Xiamen, Fujian, China

## ABSTRACT

The prospective study including 166 participants aims to evaluate the association between seminal prosaposin and the outcomes of in vitro fertilization (IVF) cycles in humans. The generalized linear model (GLM) was used to analyze the associations between seminal prosaposin concentrations and normal fertilization rates and good embryos proportion. The generalized estimating equation (GEE) was used to evaluate the association between embryo parameters and the prosaposin concentrations. Each model was adjusted for age of the couples, female basal FSH, AFC and BMI, starting dose and oocyte yield of IVF cycles and smoker. GLM models suggested that prosaposin was significantly associated with fertilization rate ($P = 0.005$) and good embryo proportion ($P = 0.038$) while none of the semen parameters (sperm concentration, motility, progressive motility, normal morphology rate, postwash sperm concentration and motility) was significantly associated with the parameters in the cohort. Using GEE, it was also shown that prosaposin was positively associated with the occurrence of early cleavage and negatively associated with uneven cleavage pattern on day 3. In both the overall population and the normozoospermia patients, the prosaposin was significantly associated with pregnancy with adjustment with covariates. In conclusion, our data suggested that seminal prosaposin concentration could provide more information regarding normal fertilization and embryo development in IVF than traditional semen parameters.

# INTRODUCTION

Semen quality, as measured according to a widely criteria established by World Health Organization (WHO), is perhaps the most important marker for male infertility potential in clinical use at the point of time (*Sakkas et al., 2015*; *WHO, 2010*). The sperm number and motility are not only the key determining factors for the "numbers game" to achieve the necessary tens to hundreds of sperm in the ampulla (*Sakkas et al., 2015*) but also the indicators for the health status of male reproductive system itself (*Choy & Eisenberg, 2018*).

Corresponding author
Jiali Cai, jialicai@xmu.edu.cn

In the era of ART, in vitro manipulation of gametes allows bypassing of the natural selection barrier for the spermatozoon in the female reproductive tract, such as hostile vaginal pH, barrier of cervical mucus and immune response in the tract, and thus reduces the importance of the "numbers game" to reach the oocytes (*Sakkas et al., 2015*). Conflicting results are obtained from studies regarding the association between male semen quality and IVF outcomes. A recent study suggested that advancing male age, elevated BMI or poor sperm quality is not associated with outcomes in frozen donor oocyte IVF cycles (*Capelouto et al., 2018*). It suggested that semen quality alone may not yield sufficient information regarding the potential of male fertility as soon as the spermatozoon reaches the oocytes.

Cumulative studies have investigated and evaluated various additional seminal/sperm markers beyond traditional seminal quality, among which the abundant tissue-specific proteins within the seminal plasma provide a rich source of potential candidates (*Bieniek, Drabovich & Lo, 2016*; *Cao et al., 2018*). Proteomic and biomarker discovery technologies have linked lists of proteins to male infertility etiologies, exposure and life styles (*Intasqui et al., 2015*). However, few proteins among the lists has been associated with the functions of the reproductive process, such as fertilization event and subsequent embryo development.

Prosaposin is known as a lysosomal protein found in Sertoli cells and the lumen of the seminiferous tubules and epididymis of mammals (*Morales et al., 1998*) as well as a secretary protein identified in the seminal proteome for both men and animals (*Codognoto et al., 2018*; *Sharma et al., 2013*; *Viana et al., 2018*). In bulls, fertility rank is positively associated seminal concentration of prosaposin (*Viana et al., 2018*). In vitro studies showed that the protein contributes to the sperm-oocyte binding, fertilization and embryo development in several species (*Amann, Hammerstedt & Shabanowitz, 1999a*; *Amann, Seidel Jr & Brink, 1999b*; *Amann et al., 1999c*; *Hammerstedt et al., 2001*; *Magargee, Cramer & Hammerstedt, 2000*). The physiological role of the protein may suggest a functional link between seminal proteins and reproductive outcomes

IVF cycles may provide an ideal model to observe the association between postulated markers and events following sperm-oocyte interaction, such as fertilization and embryo cleavage. The present study aims to evaluate the association between seminal prosaposin concentration and normal fertilization. Additionally, embryo cleavage patterns during in vitro culture are used as secondary outcomes.

## MATERIALS AND METHODS

Institutional review board approval for this study was obtained from the Ethical Committee of Medical College Xiamen University. All the subjects enrolled in this study were given written formal consent before participation. No clinical trial registry was necessary because the study did not involve any type of intervention.

### Participants

Participants were recruited between Jan 2013 and June 2016. To minimize the confounding from female participants, we included only patients receiving conventional IVF treatment for the first time, undergoing conventional long agonist for ovarian stimulation. Female participants were with good physical and mental health, aged <35 years; regular menstrual

cycles ranging from 25 to 35 days; BMI $< 28$ kg/m$^2$; normal basal serum FSH ($\leq$10 mIU/ml) and estradiol (E$_2$) ($\leq$75 pg/ml). No sign of male reproductive tract infection was detected in the male participants. The exclusion criteria were: patients with endometriosis or PCOS, patients with suboptimal ovarian response (oocyte yield <5) and patients with sign of OHSS.

## Ovarian stimulation

All patients received the same regimen using depot GnRH agonist (*Ren et al., 2014*). Patients received 2–3 ampoules (150–225 IU) gonadotrophin per day during the gonadotrophin stimulation. The starting dose was adjusted according to patients' age, AFC and BMI. Physicians triggered oocyte maturation using 5000–10000 IU human chorionic gonadotrophin (hCG; Lizhu Pharma, China) as soon as ultrasonography revealed at least one follicle measuring $\geq$18 mm in mean diameter. Oocyte retrieval was scheduled 34 to 36 hr after triggering.

## Semen preparation

Semen parameters of male counterparts were evaluated according to WHO criteria.

On the day of oocyte retrieval, semen was produced by masturbation and motile spermatozoa were prepared by centrifugal fractionation (350G, 10 min) using sperm isolation medium (Isolate, Irvine Scientific, CA). Resulting spermatozoa was washed (250G, 5min) in gamete buffer (K-SIGB-50, Cook, Australia) and incubated in 37 °C until insemination.

Semen samples (100 $\mu$l) for prosaposin determination were collected before centrifugal fractionation. The spermatozoa were removed following high speed centrifugation (14000G, 10min). The seminal plasma was stored at $-80$ °C until use.

## Embryo culture and assessment

Oocytes were inseminated 4 hr after collection. Pronuclei (PN) were identified 18 hr later. All embryos were cultured in traditional incubators (C200, Labotech, Germany) at 37 °C, 6%CO$_2$, 5%O$_2$. Occurrence of early cleavage event was observed at 27 hr post insemination. Day 3 embryos were graded based on numbers of embryo blastomere, fragmentation, and symmetry. Grade 1 and grade 2 embryos were considered as high quality embryos, grade 1, grade 2, and grade 3 embryos were considered as available embryos. Embryo transfer procedure and pregnancy determinant were described in our previous research.

## Determination of prosaposin

Prosaposin was determined using an ELISA kit (Uscn Life Science Inc., Wuhan, China) according to the manufacturer's instructions. All samples were thawed and diluted five folds in PBS. The color change of the substrate is measured spectrophotometrically at a wavelength of 450 nm. The concentration of prosaposin in the samples is then determined by comparing the O.D. of the samples to the standard curve. The lower detection limit of the analysis is less than 8.2 pg/ml.

## Statistical analysis

The primary outcome of the study was normal fertilization rate. Normal fertilization rate rather than total fertilization rate was used because it is a key performance indicator for IVF procedure (*ESHRE Special Interest Group of Embryology and Alpha Scientists in Reproductive Medicine, 2017*) and is more relevant to the outcome. The secondary outcomes were the parameters associated with embryo development following fertilization, including good embryo proportion, occurrence of early event on day 1, fragmentation, cleavage speed and cleavage pattern on day 3. Normal fertilization rate was defined as the proportion of 2PN or 2PB of cumulus-oocyte complexes inseminated (*ESHRE Special Interest Group of Embryology and Alpha Scientists in Reproductive Medicine, 2017*). Good embryo proportion was the proportion of good embryos among normally fertilized oocytes.

Generalized linear model was used to evaluate the association between prosaposin concentration and good embryos proportion. The model was adjusted for age of the couples, female basal FSH, AFC and BMI, starting dose and oocyte yield of IVF cycles and male smoker.

Generalized estimating equation was used to analyze the association between prosaposin concentrations and embryo parameters (fragmentation >10%, early cleavage, fast cleavage, slow cleavage, on time cleavage, unsynchronized cleavage and uneven cleavage). Embryos from the same cycle were treated as dependent samples in the analysis.

Traditional semen parameters (sperm concentration, motility, progressive motility, normal morphology rate, postwash sperm concentration and motility) were also associated with the outcomes in the same manner.

Multivariate model was also used to detect the association between prosaposin and pregnancy. Beside the covariate aforementioned, the model was also adjusted for endometrial thickness, stage of embryo transferred (cleavage vs blastocyst), number of embryos transferred and distance from transfer catheter tip to fundal.

Correlation between prosaposin concentrations and semen parameters were calculated according to spearmen correlation coefficient.

Data analysis was performed using SPSS Version 19.0 statistical software (IBM, Armonk, NY).

## RESULTS

A total of 166 couples were involved in the study. As demonstrated in Table 1, female counterparts were normal responders with a median oocyte yield of 11. The median age of the female patient was 31 years. None of the selected patients was with endometriosis, PCOS or other diagnosed endocrine dysfunction. Among male counterparts, the median age was 32 years. More than half of them ($n = 87$, 52.4%) were with normozoospermia according to WHO criteria. Others were with asthenozoospermia ($n = 54$), teratozoospermia ($n = 12$), asthenoteratozoospermia ($n = 9$), oligoasthenozoospermia ($n = 2$), oligoasthenoteratozoospermia ($n = 1$) or oligozoospermia($n = 1$). The median prosaposin concentration in the seminal plasma was 73.12 ng/ml. The prosaposin concentration significantly correlated with progressive

**Table 1 Patient characteristics and cycle parameters.**

| | |
|---|---|
| *n* | 166 |
| Female characteristics | |
| Female age, year | 31[5] |
| BMI, kg/m$^2$ | 21.3[3.23] |
| Basal FSH, mIU/ml | 6.66[2.17] |
| Basal LH, mIU/ml | 4.03[2.12] |
| Basal E2, pg/ml | 35[25.25] |
| AFC | 12[7] |
| Male characteristics | |
| Male age, year | 32[6.25] |
| BMI, kg/m$^2$ | 23.51[4.36] |
| Smoker (%) | 94(56.6) |
| Prosaposin, ng/ml | 73.12[95.52] |
| Semen volume, ml | 2.5[1] |
| FSH, mIU/ml | 5.15[1.91] |
| Sperm concentration, $\times 10^6$/ml | 59.69[56.2] |
| Total sperm count, $\times 10^6$ | 149.96[154.16] |
| Normal morphology, % | 6.5[6] |
| Motility, % | 50.35[22.37] |
| Total motile sperm, $\times 10^6$ | 68.96[84.13] |
| Progressive, % | 36.74[18] |
| Non progressive, % | 9[7.02] |
| Immotile, % | 49.15[22.06] |
| Postwash sperm concentration, $\times 10^6$/ml | 40[15] |
| Postwash sperm motility, % | 98[3] |
| Postwash progressive motility, % | 95[5] |
| Postwash nonprogressive motility, % | 3[2] |
| Cycle parameters | |
| Gonadotropin dose, IU | 2475[740.63] |
| Duration of stimulation, day | 12[3] |
| Starting dose, IU | 225[75] |
| Oocyte yield | 11[4] |
| Endometrial thickness, mm | 10.9[3.45] |
| Fertilization rate, % | 87.87[22.22] |
| Normal fertilization rate,% | 70[22.86] |
| Normal fertilization <50% (%) | 29 (17.5) |
| Good embryo proportion,% | 63.07[35.12] |
| ET cancelled (%) | 7 (4.2) |
| Cleavage ET (%) | 111 (69.8) |
| Blastocyst ET (%) | 48 (30.2) |
| Number of embryos transferred | 2 [1] |
| Pregnancy/ET (%) | 103/159 (64.78) |

**Notes.**
ET, embryo transfer.
Data are median [IQR] or count (percentage).

motility and postwash sperm motility but not sperm concentration or morphology (Table 2).

In multivariate analyses, the concentrations of seminal prosaposin were positively associated with normal fertilization rate ($P < 0.01$). The regression coefficient indicated that per ng/ml increase in seminal prosaposin would lead to 0.066% increase in normal fertilization rate after adjustment of confounding factors. On the other hand, however, none of the traditional semen parameters, such as sperm concentration, motility and normal morphology rate was significantly associated with normal fertilization rate (Table 3). To test the robustness of the association, we carried out a sensitivity analysis in a subgroup of patients with normozoospermia ($n = 87$). Although the $P$ value (0.042) was increased due to the small sample size, the result was consistent with that in the total population (Table 3).

To compare the ability of seminal prosaposin and standard semen parameters in predicting low fertilization events (normal fertilization rate <50%), receiver operating characteristic (ROC) curves was used to determine which cutoff would provide the best trade-off between sensitivity and specificity and AUC for each predictors were qualified (Table 4). It is shown that in both the overall population and the normozoospermia men, prosaposin showed limited discriminating capacity to predict the cycles with low fertilization, where either the sperm concentration nor the sperm motility showed any discriminating capacity (AUC<0.6, $P > 0.05$).

In Table 5, we further associated individual embryo parameter with prosaposin using GEE and found that prosaposin concentration was significantly associated with the occurrence of early cleavage on day 1 and uneven cleavage pattern on day3 (Table 5). On the other hand, the sperm motility was significantly associated with early cleavage and postwash sperm motility was negatively associated with fast cleaving of day 3 embryos.

We also explored the association between seminal prosaposin and pregnancy following embryo transfer (Table 6). With adjustment of semen volume, the prosaposin concentration was significantly associated with pregnancy. In univariate model, however, prosaposin was not significantly associated with pregnancy in either overall population (OR 1.004, 95%CI [0.998–1.004]) or Normozoospermic men (OR 1.006, 95%CI [0.998–1.014]).

## DISCUSSION

Although the detail is not clear, the role of prosaposin in male fertilization and spermatozoa-oocyte interaction has been revealed in several species (*Amann, Hammerstedt & Shabanowitz, 1999a*; *Amann, Seidel Jr & Brink, 1999b*; *Amann et al., 1999c*; *Hammerstedt et al., 2001*; *Magargee, Cramer & Hammerstedt, 2000*). Conservation of evolution may suggest the importance of the protein in male reproductive process. However, it is still not known whether there is any clinical importance of this protein among patients receiving infertility treatment. The present study is adding to existing knowledge by demonstrating the role of seminal prosaposin in predicting the fertilization, embryo development and pregnancy in patients receiving IVF. The data suggested that prosaposin concentration in semen not only had moderate predicting value in low fertilization event but also significantly

**Table 2  Correlation between prosaposin concentration and semen parameters.**

|  | Correlation coefficient | P |
|---|---|---|
| Sperm concentration | 0.13 | 0.096 |
| Normal morphology | −0.153 | 0.051 |
| Motility | −0.01 | 0.893 |
| Progressive motility | 0.197 | 0.011 |
| Postwash sperm concentration | −0.115 | 0.141 |
| Postwash sperm motility | 0.269 | <0.001 |
| Postwash progressive motility | 0.266 | 0.001 |
| FSH levels | 0.098 | 0.21 |

correlated to embryo development and pregnancy following fertilization. Because seminal plasma is removed during preparation of spermatozoon in IVF procedures, the prosaposin in seminal plasma may not be directly involved in the sperm-oocyte interaction. It is possible that the concentration of prosaposin reflects the health status of reproductive system. As shown in our previous study, the prosaposin associated with spermatozoon may reflect the external exposure and internal body burden of environmental pollutants (*Cai et al., 2015*). The finding also echoes a recent proteomics study in which seminal proteins such as prosaposin may indicate the fertility ranking of the males in bulls (*Viana et al., 2018*).

A number of research groups have associated seminal plasma protein levels with semen parameters (*Cao et al., 2018*; *Davalieva et al., 2012*; *Diamandis et al., 1999*; *Drabovich et al., 2013*; *Freour et al., 2013*; *Rolland et al., 2013*; *Wang et al., 2009*). Panels of candidate proteins for male fertility have been proposed in studies comparing the proteomics data between normozoospermic men and men with asthenozoospermia, oligozoospermia, or azoospermia. The semen parameters, however, may only have limited predicting value for clinic outcomes among IVF patients. Bartolacci et al. suggested that oligozoospermia according to WHO criteria may affect the embryo development but not top quality blastocyst formation rate or the establishment of pregnancy in ICSI cycles (*Bartolacci et al., 2018*). In a study including 1280 IVF cycles, Mariappen et al. suggested that the semen parameters have an insignificant role to play in embryo quality and overall outcomes (*Mariappen et al., 2018*). Similarly, Capelouto et al. found a lack of association between semen parameters and live birth rate in frozen donor oocyte cycles (*Capelouto et al., 2018*). These studies may lie on the fact that several critical processes of natural conception are bypassed by the ART treatment and suggest that male fertility biomarkers screened according to semen parameters may not be as feasible as they were expected in ART populations. In the present study, the association between seminal protein and outcomes of IVF treatment was not only observed in the overall population but also in normozoospermic men. On the other hand, neither sperm concentration nor sperm motility was significantly associated with normal fertilization and pregnancy. The data supported the hypothesis that seminal proteins provide more information regarding fertility than routine semen analysis and suggested that male factor might still affect the reproductive outcome even though the semen parameters are considered as normal according to the established criteria.

**Table 3** **Multivariate analysis for normal fertilization rate and good embryo proportion, with respect to prosaposin concentration and semen quality.** Each model was adjusted for age of the couple, female BMI, female basal FSH, AFC, starting dose, oocyte yield and male smoker.

| | Normal fertilization | | Good embryo proportion | |
|---|---|---|---|---|
| | Coefficient | P | Coefficient | P |
| Overall population, $n = 166$ | | | | |
| Prosaposin | 0.066 | 0.005 | 0.070 | 0.038 |
| Sperm concentration | 0.046 | 0.118 | −0.040 | 0.342 |
| Male FSH | 0.169 | 0.801 | −1.984 | 0.033 |
| Normal morphology | 0.115 | 0.685 | 0.100 | 0.800 |
| Motility | 0.007 | 0.934 | −0.203 | 0.422 |
| Progressive motility | 0.067 | 0.499 | 0.067 | 0.632 |
| Postwash sperm concentration | −0.071 | 0.554 | −0.273 | 0.101 |
| Postwash motility | 0.005 | 0.979 | −0.234 | 0.358 |
| Postwash progressive motility | 0.077 | 0.664 | −0.210 | 0.399 |
| Normozoospermia, $n = 87$ | | | | |
| Prosaposin | 0.066 | 0.042 | 0.049 | 0.325 |
| Sperm concentration | 0.045 | 0.330 | 0.006 | 0.925 |
| Male FSH | −0.248 | 0.781 | −2.3 | 0.066 |
| Normal morphology | 0.233 | 0.577 | 0.314 | 0.595 |
| Motility | −0.038 | 0.837 | −0.059 | 0.819 |
| Progressive motility | 0.026 | 0.921 | 0.244 | 0.511 |
| Postwash sperm concentration | 0.194 | 0.296 | −0.479 | 0.066 |
| Postwash motility | 0.855 | 0.586 | 4.681 | 0.032 |
| Postwash progressive motility | 0.843 | 0.361 | 1.940 | 0.135 |

Our data also provide a detailed look at the association male factor and embryo development by linking the individual embryo morphological parameters with semen parameters and seminal protein using GEE. As shown in Table 5, male factors have been associated with cleavage events at early development, such as early cleavage on day 1 and cleavage rates on day 3. Occurrence of early cleavage at a given time point is deemed as a significant predictor for embryo implantation (*Alpha Scientists in Reproductive Medicine and ESHRE Special Interest Group of Embryology, 2011*). The use of modern time lapse technology further confirmed the importance of the timing and period of early cleavages before genomic activation in predicting embryo developmental competence (*Kaser et al., 2017*). Under experimental condition, it is shown that the 2nd to 3rd mitoses were sensitive periods in the presence of spermatozoal oxidative stress (*Burruel et al., 2014*). It could also be postulated that early cleavage events of embryos were also sensitive to spermatozoal oxidative stress derived from physiological or pathological conditions.

Due to the complexity of reproductive process, it is difficult to associate a single semen maker with pregnancy outcome. Conflicting evidence regarding the association between male factors and ART outcomes was not only observed in studied using semen parameters as male fertility markers (*Bartolacci et al., 2018*; *Borges Jr et al., 2016*; *Capelouto et al., 2018*; *Chapuis et al., 2017*; *Mariappen et al., 2018*; *Mazzilli et al., 2017*), but also in those using

**Table 4  Discriminating capacity of prosaposin concentration and semen quality for low fertilization (normal fertilization < 50%) in ROC curves.**

|  | Cutoff | Sensitivity | Specificity | AUC (95%CI) | P |
|---|---|---|---|---|---|
| Overall population, $n = 166$ |  |  |  |  |  |
| Prosaposin | 63.35 | 0.76 | 0.60 | 0.666(0.569–0.763) | 0.006 |
| Sperm concentration | 48.63 | 0.55 | 0.67 | 0.582(0.46–0.705) | 0.171 |
| FSH | 3.39 | 0.86 | 0.21 | 0.474(0.362–0.585) | 0.655 |
| Normal morphology | 17.40 | 1.00 | 0.07 | 0.433(0.325–0.541) | 0.266 |
| Motility | 63.53 | 0.93 | 0.15 | 0.511(0.389–0.632) | 0.861 |
| Progressive motility | 36.74 | 0.59 | 0.53 | 0.548(0.426–0.67) | 0.429 |
| Postwash sperm concentration | 52.50 | 0.90 | 0.19 | 0.529(0.419–0.638) | 0.632 |
| Postwash motility | 96.50 | 0.48 | 0.72 | 0.604(0.483–0.725) | 0.085 |
| Postwash progressive motility | 91.00 | 0.48 | 0.69 | 0.602(0.485–0.719) | 0.089 |
| Normozoospermia, $n = 87$ |  |  |  |  |  |
| Prosaposin | 63.35 | 0.88 | 0.62 | 0.707(0.592–0.822) | 0.010 |
| Sperm concentration | 48.63 | 0.50 | 0.77 | 0.61(0.456–0.764) | 0.171 |
| FSH | 8.81 | 0.063 | 0.972 | 0.436(0.284–0.588) | 0.436 |
| Normal morphology | 11.75 | 0.88 | 0.25 | 0.437(0.294–0.581) | 0.437 |
| Motility | 63.53 | 0.88 | 0.21 | 0.412(0.248–0.576) | 0.276 |
| Progressive motility | 36.74 | 0.25 | 0.87 | 0.516(0.346–0.686) | 0.844 |
| Postwash sperm concentration | 31.50 | 0.38 | 0.86 | 0.614(0.446–0.781) | 0.158 |
| Postwash motility | 96.00 | 0.38 | 0.82 | 0.547(0.372–0.722) | 0.558 |
| Postwash progressive motility | 91.00 | 0.38 | 0.79 | 0.573(0.414–0.732) | 0.363 |

other male fertility biomarkers, such as DNA fragmentation (*Colaco & Sakkas, 2018*). While many of the previous analyses were univariate in nature, the conflicting results may imply the importance of adjustment for the confounding factors associated with maternal factors. During the reproductive process, the female contribute to not only the maternal genetic material but also most of the cell machinery of the zygote and the environment for embryo implantation and fetal growth. In predicting the outcomes of IVF, maternal factors such as gynecological etiologies, ovarian response, maternal age and endometrial thickness may play significant roles (*McLernon et al., 2016*). In their study investigating the effects of male factor on ART outcomes, Mariappen et al. found that female age but not male age or semen parameters has significant influence on pregnancy or live birth (*Mariappen et al., 2018*). In our study, the association between male fertility and IVF outcomes is strengthened by a prospective cohort in which female counterparts with good prognosis were selected and important covariates for IVF outcomes were considered.

## CONCLUSIONS

In conclusion, the present study demonstrated the association between prosaposin, a seminal secretory protein and the occurrence of fertilization and embryo cleavage events. Our data also suggested that seminal proteins may provide more information regarding IVF outcomes than traditional semen parameter could yield. Although the AUC suggest only a limited discriminating capacity of prosaposin to predict low fertilization events, a

**Table 5  Association between embryo parameters and prosaposin concentration/semen quality.** Each model was adjusted for age of the couple, female BMI, basal FSH, AFC, starting dose, oocyte yield and male smoker. Fast cleaving embryo was defined as an embryo with more than eight cells at the time of observation; slow cleaving embryo was defined as an embryo with less than eight cells at the time of observation; Nonsynchronized cleaving embryo was defined as an embryo with even cell number at the time of observation.

| | Early cleavage on day 1 | Fast cleaving embryo on day3 | Slow cleaving embryo on day3 | On time 8 cell embryo on day3 | Unsynchronized cleaving embryo on day3 | Fragmentation>10% on day3 | Uneven cleaving embryo on day3 |
|---|---|---|---|---|---|---|---|
| **Overall population, $n = 1252$** | | | | | | | |
| Prosaposin | 1.005(1.002–1.009)** | 1.002(0.999–1.005) | 0.999(0.996–1.002) | 0.999(0.997–1.002) | 1.001(0.999–1.003) | 0.997(0.993–1.001) | 0.997(0.995–0.999)* |
| Sperm concentration | 1.001(0.997–1.005) | 0.998(0.994–1.002) | 1.002(0.997–1.006) | 1.001(0.997–1.003) | 1.001(0.998–1.002) | 1.001(0.995–1.004) | 1.001(0.998–1.004) |
| Male FSH | 0.962(0.89–1.04) | 1.051(0.98–1.128) | 0.99(0.94–1.045) | 0.969(0.93–1.014) | 0.978(0.93–1.024) | 1.043(0.96–1.137) | 1.07(1.01–1.139)* |
| Normal morphology | 0.994(0.952–1.038) | 0.995(0.963–1.029) | 0.992(0.966–1.018) | 1.015(0.993–1.037) | 0.997(0.977–1.017) | 0.981(0.933–1.033) | 0.984(0.952–1.016) |
| Motility | 0.999(0.986–1.011) | 0.998(0.979–1.017) | 0.994(0.98–1.008) | 1.002(0.989–1.016) | 0.999(0.992–1.006) | 0.96(0.876–1.053) | 1.011(0.996–1.025) |
| Progressive motility | 1.020(1.005-1.036)** | 0.992(0.981–1.003) | 1.001(0.988–1.012) | 1.002(0.992–1.012) | 1.003(0.993–1.013) | 1.012(0.994–1.030) | 1.003(0.992–1.014) |
| Postwash sperm concentration | 0.997(0.982–1.013) | 1.001(0.987–1.014) | 1.003(0.992–1.014) | 0.996(0.986–1.006) | 1.007(0.999–1.016) | 1.015(0.953–1.081) | 1.003(0.986–1.021) |
| Postwash motility | 1.087(0.960–1.232) | 0.982(0.974–0.991) ** | 1.007(0.999–1.015) | 1.012(0.994–1.030) | 1.005(0.998–1.013) | 1.016(0.954–1.081) | 1.020(0.984–1.058) |
| Postwash progressive motility | 1.028(0.983–1.076) | 0.980(0.968–0.991)** | 1.010(0.999–1.020) | 1.010(0.994–1.026) | 1.003(0.993–1.014) | 1.017(0.955–1.082) | 1.008(0.984–1.033) |
| **Normozoospermia, $n = 645$** | | | | | | | |
| Prosaposin | 1.003(0.999–1.007) | 1.004(1.001–1.007)** | 0.996(0.993–0.999)* | 1.001(0.998–1.002) | 1.001(0.998–1.003) | 0.997(0.991–1.003) | 1.001(0.997–1.003) |
| Sperm concentration | 1.003(0.998–1.008) | 1(0.996–1.005) | 0.999(0.995–1.003) | 1.002(0.999–1.006) | 0.998(0.994–1.002) | 0.996(0.987–1.005) | 0.998(0.992–1.003) |
| Male FSH | 0.936(0.88–0.996)* | 0.96(0.89–1.039) | 1.061(1–1.127) | 0.964(0.92–1.009) | 0.996(0.95–1.05) | 1.077(0.94–1.23) | 1(0.94–1.059) |
| Normal morphology | 0.998(0.933–1.067) | 1.024(0.974–1.076) | 0.99(0.948–1.034) | 1.003(0.974–1.033) | 1.019(0.99–1.049) | 0.939(0.88–1.001) | 0.999(0.969–1.03) |
| Motility | 0.995(0.974–1.017) | 1.009(0.991–1.028) | 0.987(0.968–1.006) | 1.007(0.993–1.022) | 0.987(0.969–1.004) | 0.98(0.945–1.017) | 0.985(0.966–1.004) |
| Progressive motility | 1.019(0.991–1.048) | 0.992(0.968–1.017) | 0.983(0.959–1.006) | 1.021(1.003–1.039)* | 0.995(0.973–1.017) | 1.003(0.963–1.045) | 0.991(0.966–1.018) |
| Postwash sperm concentration | 1.003(0.982–1.025) | 1.001(0.98–1.022) | 1.019(0.998–1.04) | 0.984(0.97–0.998)* | 1.016(1.00–1.033)* | 1.031(0.997–1.066) | 1.004(0.98–1.029) |
| Postwash motility | 1.017(0.811–1.275) | 1.022(0.847–1.234) | 0.935(0.793–1.101) | 1.062(0.924–1.22) | 0.943(0.831–1.07) | 0.896(0.696–1.155) | 0.978(0.831–1.152) |
| Postwash progressive motility | 0.968(0.846–1.107) | 0.985(0.883–1.099) | 1.011(0.914–1.119) | 1.011(0.934–1.093) | 0.977(0.906–1.053) | 0.949(0.82–1.098) | 0.966(0.878–1.064) |

**Notes.**

*Indicates significant at $P < 0.05$.

Table 6  Multivariate analysis forclinical pregnancy ($n = 157$).

| Variable | Category | OR (95% CI) | |
| --- | --- | --- | --- |
| | | Overall | Normozoospermia |
| Female age | per year increased | 0.973(0.834–1.135) | 0.905(0.704–1.162) |
| Male age | per year increased | 1.028(0.899–1.175) | 1.095(0.885–1.355) |
| Female BMI | per unit increased | 0.982(0.859–1.122) | 0.917(0.768–1.094) |
| Female basal FSH | per mIU/ml increased | 0.985(0.802–1.211) | 1.559(0.992–2.451) |
| Male FSH | per mIU/ml increased | 1.06(0.877–1.281) | 1.041(0.718–1.507) |
| AFC | per AFC increased | 1.008(0.923–1.101) | 1.067(0.933–1.219) |
| Male smoker | no smoker vs smoker | 0.468(0.206–1.063) | 0.597(0.158–2.249) |
| Prosaposin concentration | per ng/ml increased | 1.009(1.001–1.016)[*] | 1.018(1.004–1.032)[*] |
| Semen volume | per ml increased | 1.687(1.059–2.687)[*] | 3.3(1.448–7.52)[*] |
| Sperm concentration | per $10^6$/ml increased | 0.994(0.986–1.003) | 0.991(0.976–1.006) |
| Normal morphology | per percentage increased | 1.019(0.935–1.11) | 1.258(1.048–1.511)[*] |
| Motility | per percentage increased | 1.007(0.984–1.031) | 1.029(0.973–1.087) |
| Starting dose | per IU increased | 1.002(0.991–1.015) | 1.01(0.991–1.029) |
| Oocyte yield | per oocyte increased | 1.094(0.956–1.252) | 1.121(0.881–1.426) |
| Distance to fundal | per cm increased | 0.953(0.308–2.953) | 0.303(0.046–1.985) |
| Number of embryo transferred | per embryo increased | 0.697(0.219–2.219) | 4.761(0.452–50.106) |
| Endometrial thickness | per mm increased | 1.244(1.036–1.493)[*] | 1.221(0.932–1.6) |
| At least on top quality embryo transferred | yes vs no | 5.541(1.436–21.381)[*] | 35.2(2.846–435.317)[*] |
| Stage of embryo transfer | cleavage vs blastocyst | 1.158(0.334–4.021) | 8.031(0.704–91.582) |

Notes.
*Indicates significant at $P < 0.05$.

panel of seminal proteomics markers may provide a higher discriminating capacity in the future.

## ACKNOWLEDGEMENTS

The authors thank all the staff, especially the embryologists in our lab for their support in generating this manuscript. We would like to thank Xinli Wang for her assistance in data processing.

### Funding

This work was supported by the National Natural Science Foundation of China (81302454), the Natural Science Foundation of Fujian Province (2016D025), the Special Fund for Clinical and Scientific Research of Chinese Medical Association (18010360765) and the Xiamen medical advantage subspecialty construction project (2018296). There was no additional external funding received for this study. The funders had no role in study design, data collection and analysis, decision to publish, or preparation of the manuscript.

### Grant Disclosures

The following grant information was disclosed by the authors:

National Natural Science Foundation of China: 81302454.
Natural Science Foundation of Fujian Province: 2016D025.
Special Fund for Clinical and Scientific Research of Chinese Medical Association: 18010360765.
Xiamen medical advantage subspecialty construction project: 2018296.

## Competing Interests

The authors declare there are no competing interests.

## Author Contributions

- Chun Xu conceived and designed the experiments, performed the experiments, analyzed the data, prepared figures and/or tables, authored or reviewed drafts of the paper, approved the final draft.
- Jiali Cai conceived and designed the experiments, performed the experiments, analyzed the data, contributed reagents/materials/analysis tools, prepared figures and/or tables, authored or reviewed drafts of the paper, approved the final draft.
- Lanlan Liu conceived and designed the experiments, contributed reagents/materials/-analysis tools, prepared figures and/or tables, authored or reviewed drafts of the paper, approved the final draft.
- Jianzhi Ren conceived and designed the experiments, analyzed the data, contributed reagents/materials/analysis tools, authored or reviewed drafts of the paper, approved the final draft.

## Human Ethics

The following information was supplied relating to ethical approvals (i.e., approving body and any reference numbers):

The Institutional Review Board at Chinese PLA 174th Hospital approved this reasearch.

## Data Availability

The raw measurements are available in the Supplemental Files.

## Supplemental Information

Supplemental information for this article can be found online at http://dx.doi.org/10.7717/peerj.8177#supplemental-information.

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
