# Peer review of "Prosaposin in seminal plasma on the day of oocyte retrieval is associated with normal fertilization and embryo development in in vitro fertilization cycles"

_PeerJ, doi:10.7717/peerj.8177_

## Round 0.1 · original submission · Minor Revisions

Our reviewers are enthusiastic about your paper and have some suggestions for areas where it can be improved. I look forward to receiving an updated version!

·

Basic reporting

I found a typo in line 113, "grad" was "grade".

Experimental design

The authors said "The primary outcome of the study was normal fertilization rate" in line 123. Do we need the word "normal" here?
I think just "fertilization rate" is enough.

Validity of the findings

I could not understand the statement in Results section, line 179-180 "With adjustment of semen volume, the prosaposin concentration was significantly associated with pregnancy". Where I should refer the data in Table 6? And in Table 6, I could not find the mark "*" indicates significant at P<0.05. The authors should make clear the basis for making that statement.

The discussion is lengthy and not clear, so why don't the authors summarize it a little shorter? For example, the fourth paragraph is able to combine to the first paragraph about the nature of prosaposin.
In the line 251, the authors claimed "...between male fertility and ART outcomes...". However, in this study, the authors included only IVF cases. I recommend the authors to reconsider the use of the word "ART".

Additional comments

I read this manuscript with great interest. It is very difficult to find a marker in semen to predict pregnancy, because the cases of infertility are variable and complicate. Therefore, I am not surprised that the authors could not find significant correlation to fertilization. On the other hand, the authors found the correlation embryo development. I think this result mean that prosaposin supports the role of sperm function for early development of embryo.

·

Basic reporting

no comment

Experimental design

no comment

Validity of the findings

mention below

Additional comments

Dear Editor-in-chief
As you send to me a manuscript entitled " Prosaposin in seminal plasma on the day of oocyte retrieval is associated with normal fertilization and embryo development in in vitro fertilization cycles. "
I reviewed this MN, it need minor revision.
The manuscript was written well and there were little grammatical mistakes.
Because the aim of this manuscript is evaluate the level of Prosaposin in the semen, and this protein produced by sertoli cells and epithelium of epidydyme. These cells have been controlled by FSH hormone, so it is necessary to check the level of FSH hormone in male partners to check the function of these cells that contribute to synthesis of Prosaposin, while the authors check the FSH hormone in female partners.

---

## Round 0.2 · accepted · Accept

Thank you for addressing the reviewers' queries.

·

Basic reporting

no comment

Experimental design

no comment

Validity of the findings

no comment

Additional comments

no comment

·

Basic reporting

the manuscript is written well and is an applied one.

Experimental design

well.

Validity of the findings

the results that obtained in this manuscript are novel and explain the hypothesis.

Additional comments

the manuscript entitled" Prosaposin in seminal plasma on the day of oocyte retrieval is
associated with normal fertilization and embryo development in in vitro fertilization cycles". was reviewed. the authors justify the items that informed before and improve it . so it seems the manuscript is suitable for publishing.